# Introducing *zanadio*—A Digitalized, Multimodal Program to Treat Obesity

**DOI:** 10.3390/nu14153172

**Published:** 2022-08-01

**Authors:** Katarina Forkmann, Lena Roth, Nora Mehl

**Affiliations:** 1aidhere GmbH, 20354 Hamburg, Germany; nora.mehl@aidhere.de; 2Max Planck Institute for Human Cognitive and Brain Sciences, 04103 Leipzig, Germany; rothl@cbs.mpg.de

**Keywords:** *zanadio*, digital weight reduction program, multimodal obesity treatment, mHealth, digital disease management, digitalization, digital health application

## Abstract

While the prevalence of overweight and obesity has been increasing annually, the accessibility of on-site treatment programs is not rising correspondingly. Digital, evidence-based obesity treatment programs could potentially alleviate this situation. The application *zanadio* has been developed to enable patients with obesity (BMI 30–45 kg/m^2^) to participate in a digital, multimodal weight reduction program based on current treatment guidelines. This article is divided into two parts: (I) it introduces *zanadio*, its aims and therapeutic concept, and (II) provides a first impression and demographic data on more than 11,000 patients from across the country who have used *zanadio* within the last 16 months, which demonstrates the demand for a digital obesity treatment. *zanadio* has the potential to partially close the current gap in obesity care. Future work should focus on identifying predictors of successful weight loss to further individualize digital obesity treatment, and an important next step would be to prevent obesity, i.e., to start the treatment at lower BMI levels, and to invent digital treatment programs for children and adolescents.

## 1. Introduction

Obesity is a rapidly growing health-care problem in most parts of the world. Projections show that the prevalence of obesity will continuously increase within the next few years [1]. In Germany, about 20–25% of adults are considered to be obese [2]. The current global pandemic situation and the actions taken by the government to contain the SARS-CoV-2 virus worsened this situation and led to a lack of physical activities, increased unhealthy eating behaviors, and psychological as well as psychosocial burdens. As a result, people’s weights increased significantly during the first months of the pandemic [3,4]. Furthermore, the access to outpatient treatment programs has been limited more than ever before. Evidence-based, digital treatment options are an alternative to on-site treatments as they can be used anywhere and anytime. Although similar initiatives exist in other countries, e.g., [5,6,7], Germany was the first country that introduced so-called digital health applications (abbreviated ‘DiGA’) in December 2019 after passing two laws, the Digital Healthcare Act (DVG) and Digital Health Applications Ordinance (DiGAV; i.e., an ordinance on the procedure and requirements for testing the eligibility for reimbursement of digital health applications in the health insurance system). The singularity of the German DiGA system lies within the full reimbursement of an approved digital treatment program by every public health insurance program, granting a high number of patients full and easy access without additional costs. Digital health applications are medical devices with low risk (risk class I or IIa according to the European legal framework for medical devices; Medical Device Regulation, MDR) that can be prescribed by physicians and psychotherapists [8]. It is a precondition, however, that these applications have been proven to have positive medical effects for the intended patient group. Thus, after fulfilling all formal and safety requirements (according to MDR), a research study must be conducted within one year after market launch to demonstrate clinically relevant and statistically significant positive healthcare effects of the DiGA.

*zanadio* is a smartphone application that has been developed to provide a digital and multimodal treatment for patients diagnosed with obesity (defined as a body mass index (BMI) of over 30 kg/m^2^). *zanadio* was one of the first digital health applications to be approved by the German Federal Institute for Drugs and Medical Devices (BfArM). It is currently approved to be used for the treatment of obesity within a BMI range of 30–45 kg/m^2^ [9]. To date, one additional DiGA that specifically aims at treating obesity is available (oviva Direkt). Other DiGAs are primarily intended to treat diabetes, but are also supporting patients regarding obesity management (e.g., Vitadio [10,11]). In addition, several multimodal, on-site treatment programs are available in Germany. However, the latter are only accessible to a small number of patients and are further temporarily and locally restricted, i.e., they cannot be used independent of time and place of residence. Moreover, they are not always reimbursed by health insurance, thereby leading to additional costs for patients. Thus, digital obesity therapies, such as *zanadio*, provide an opportunity to reach a great number of patients that either don’t have access to on-site treatment programs, won’t participate in such a program, or are on a waiting list for therapy.

The following article is divided into two parts. The first part will give a brief overview about the treatment approach implemented by *zanadio*. The second part will provide initial findings on real-world data, focusing on demographic data of patients that have been using *zanadio* within the first year of its availability.

## 2. Multimodal Treatment of Obesity: The Therapeutic Concept of *zanadio*

*zanadio* follows a multimodal treatment approach and is based on guidelines and treatment recommendations that are currently in place in Germany [12]. It aims at supporting patients to individually and beneficially change their lifestyle in order to effectively reduce their body weight, increase their health-related behavior and thus reduce the symptom severity of other diseases or their risk of developing obesity-associated diseases [12,13,14,15]. In short, behavioral changes include a reduction in calorie intake, and, at the same time, an increased demand for nutrient-dense calories due to increased physical activities, resulting in a negative caloric balance, weight loss, and loss of body fat. Importantly, the app aims to empower patients in obesity-related self-management. They will learn to independently use different behavioral techniques to develop and maintain a healthy lifestyle. To this end, validated methods from behavioral science, nutrition, and exercise therapy are used to achieve a long-term change in individual health behavior.

The program content comprises the following sub-areas:

*Knowledge transfer:* The cornerstone of behavior change is knowledge about the effects of nutrition, exercise, and behavior on weight gain and weight loss. This knowledge is conveyed via e-learning modules, which include texts, videos, or regular webinars with a specific monthly focus on one of the three pillars of the program. The acquired knowledge is regularly refreshed, e.g., through questions for personal reflection or small quizzes.

*Change*: Guided by nutrition and exercise tracking integrated into the app, patients are able to initially observe their behavior and the impact of individual choices on weight change success. This is achieved by a user-friendly and clearly arranged presentation of individual results. Patients are encouraged to define personal intermediate and final goals using the SMART technique [16] to enhance adherence by increasing motivation and avoiding frustration.

*Motivation & Support: *zanadio** includes various elements supposed to strengthen patients’ motivation to change. A chat with qualified professionals (e.g., holding degrees in human medicine, dietetics, ecotrophology, psychological psychotherapy, psychology, physiotherapy, or sports science) individually supports patients throughout the program. The qualified support can be contacted via chat whenever patients have questions regarding the content and functionality of the application. Furthermore, patients receive automated feedback, such as weekly reports, reminders, or motivational messages. In addition to the contact via the chat function, all patients receive a one-time video call with a certified dietician in order to assess the patient’s individual treatment needs and risk profile. The program can be supplemented by personal consultations, which is assumed to positively influence the effectiveness. This service, however, has to be paid for by the patient.

*zanadio* can be connected to different devices, such as weight scales and activity trackers of a variety of manufacturers, to automatically record weight and physical activities. This is recommended to assess weight changes, calorie intake, and calorie consumption due to physical activity as correctly as possible. Regular tracking allows monitoring an individuals’ change in behavior and its effect on body weight and is thus assumed to improve the effects on health, motivation, and adherence to the program [17,18]. However, using such devices within the *zanadio* program is not mandatory as all of this information can be entered manually.

The minimal recommended treatment duration of *zanadio* is 6 months. The multimodal treatment was, however, conceptualized to be used for 12 months, aiming for a beneficial and sustainable lifestyle change to reach persistent weight-loss, improved quality of life and wellbeing, as well as reductions in obesity-associated risk factors, such as type 2 diabetes, hypertension, coronary heart diseases, and a number of psychological problems [2,19,20]. For patients with obesity, national and international guidelines [12,21] recommend a sustained weight reduction of at least 5% of initial body weight as this amount of weight loss has been shown to already reduce obesity-related physical and psychological symptoms and comorbidities [14,21,22,23,24,25]. Patients who are using *zanadio* for 12 months are, on average, assumed to show a clinically relevant weight reduction of at least 5%. This weight loss goal can be expected as comparable weight reductions could be demonstrated in other multimodal on-site or digital weight reduction programs incorporating comparable treatment modules (e.g., −7.25% in [26]; −6.11 kg (corresponding to −6.45%) in [27]; −4.5 kg (corresponding to −3.6%) in [14]; −5.2% in [28]; −4% within 3 months in [10]; and −5.9% within 6 months in [11]).

*zanadio* is indicated for individuals with obesity (ICD-10 code E66) and an Edmonton Obesity Staging System (EOSS) level 0–2 [29]. Indications and prerequisites, as well as contraindications and exclusion criteria, are listed in Table 1. The program is so far primarily indicated to be used within a BMI range of 30–40 kg/m^2^ but can also be prescribed for BMI 40–45 kg/m^2^. In this case, regular medical monitoring every 3 months is recommended. In general, the program is supposed to be used by patients only, allowing them to manage their own disease and treatment. However, the manufacturer of *zanadio* recommends an initial examination and dialogue between patients and their physicians/psychotherapists when prescribing *zanadio* in order to clarify the indication as well as individual motivation and goals [30]. Further, semi-annual visits to review individual progress and to assess changes in clinical parameters (e.g., anthropometric data, blood tests, bioimpedance analysis) are recommended. Importantly, the prescription for *zanadio* (and every other digital health application) must be renewed every 3 months, allowing physicians to regularly communicate with their patients, to evaluate and discuss therapy progress and to maintain a stable patient-physician relationship. To this end, the *zanadio* app provides a report that summarizes the baseline condition and current individual therapy progress and serves as the basis for regular doctor-patient interactions.

To date, *zanadio* is preliminarily listed in the German digital health applications directory (DiGA directory). According to the so-called fast-track process for digital health applications, all DiGAs are required to demonstrate their efficacy by means of an independently conducted clinical study within one year of market launch. After demonstrating their efficacy in terms of positive care effects, i.e., medical benefit or patient-relevant improvement of structure and processes, new digital health applications will be permanently listed in the DiGA directory. Such a clinical trial has recently been conducted for *zanadio* (trial registration: German Clinical Trial Register, DRKS, DRKS00024415). The results of the randomized clinical trial will be published elsewhere. For more information on the regulatory requirements for the approval of a digital health application, please refer to the guidelines published by the German Federal Institute for Drugs and Medical Devices (BfArM) [31].

## 3. Who Is Using *zanadio*? A Brief Patient Description Using Real-World Data

Since its preliminary listing in the directory of digital health applications, *zanadio* can be used by patients with obesity upon prescription. More than 12,000 patients have since been registered. In this study, a first and concise description of the patients having registered to the health application *zanadio* between November 2020 and April 2022 will be provided. An additional analysis on the performance of *zanadio* based on real-world data will be conducted soon.

Overall, *n* = 11.323 patients with a BMI between 30 and 45 kg/m^2^ received a prescription, registered as users, and gave consent for their data to be used for scientific purposes. Although the prevalence of overweight and obesity is comparably high in men and women [13], *zanadio* is more frequently used by women (79.6% female, 20.4% male). At enrollment in the treatment program, patients are, on average, 45.6 ± 12.7 years old (M ± SD; range: 18–87 years) and have an average weight of 106.3 ± 15.6 kg (67–180 kg), corresponding to an average BMI of 36.6 ± 3.8 kg/m^2^ (obesity class I: *n* = 4750 (42.0%), obesity class II: *n* = 4766 (42.1%), obesity class III: *n* = 1807 (16.0%). The distribution of age and BMI is nearly identical for male and female patients (see Figure 1). Table 2 contains further information on patient demographics.

Looking at the comorbidity profile of patients using *zanadio*, the most frequently self-reported comorbidities were hypertension (42.3%; frequencies refer to patients for whom data on comorbidities are available), thyroid diseases (32.9%), and spine and joint diseases (30.4%). Diabetes was reported by 17.8% of patients (no differentiation between type 1 or type 2 diabetes). As is expected, the frequency of self-reported diabetes increases with higher BMI (obesity class I: 14.0%; obesity class II: 15.5%; obesity class III: 16.4%). The comorbidity profile of *zanadio* patients is highly overlapping with comorbidities reported in the general obese population [32,33,34]. Self-reported comorbidities and their respective frequencies are given in Figure 2. It has to be noted that participating in the *zanadio* program is not indicated in the case of acute untreated or unstable mental conditions (e.g., depressive episode, anxiety disorder). In order to ensure that *zanadio* is not used in the case of an acute psychiatric crisis, all patients receive a comprehensive one-time video call with a certified dietician to assess the patient’s individual treatment needs and risk profile.

In general, the patients using *zanadio* consider weight loss to be of great importance (M ± SD: 9.07 ± 1.27; median = 10 on a 1–10 scale (verbal anchors: 1 = “not important at all”, 10 = “very important”). As is well-known from outpatient care, most patients have a long history of various weight loss attempts. Before using *zanadio* as a digital treatment for obesity, patients tried on average 1.8 ± 0.9 different types of diets (M ± SD, range: 1–7; note that the online survey proposed seven different types of weight-loss programs/diets including the category “others”). Self-controlled and app-assisted diets were most frequently reported (50.6%), followed by Weight Watchers (25.6%), and formula diets (16.4%).

A major advantage compared to outpatient, multimodal treatment approaches is the independence of the program in terms of time and place of residence. The fact that a majority of *zanadio* patients work either full-time (34.4%), part-time (20.5%), or even in shifts (11.7%), and that 57.0% of patients also have children (about 50% of these children are still living at their parent’s home), highlights the importance of making it easy to integrate weight loss programs into an individual’s everyday professional and private life. Due to their flexible use, digital health applications, such as *zanadio*, are well suited to be used in a way that meets the individual needs of patients with obesity.

## 4. Discussion

There is a great discrepancy between the high and increasing prevalence of patients with obesity in countries such as Germany and the restricted availability of programs to treat obesity. There is clearly a need to provide more, and ideally different, therapeutic options that are available to the high number of patients. Digital health applications offer a new and promising way to ensure patients have immediate access to a treatment.

As one of the first digital health applications in Germany, *zanadio* aims to help patients with obesity (BMI 30–45 kg/m^2^) to reduce their excess body weight. The program follows a guideline-based, multimodal treatment approach to aid patients in achieving such long-term weight loss through sustainable changes in nutrition, physical exercise, and behavior.

As for each medical device, strict regulatory criteria have to be met for digital health applications. Further, clinically relevant and statistically significant therapeutic benefits have to be proven by means of a clinical trial. *zanadio* fulfills these regulatory criteria and an independent randomized controlled trial has been conducted to evaluate the treatment effects of *zanadio*. The final results of the clinical trial will be published in the near future.

### 4.1. Advantages of a Digital Weight Reduction Program

Analogous, long-term treatment and on-site care programs for patients with obesity bring their own challenges. Barriers include, for example, limited access to a treatment center, availability of free treatment slots, time requirements of the program, and integration into the patient’s private and professional life [35]. As a digital health application, *zanadio* can be used independent of time and place of residence, making it a low-threshold treatment option that is easily accessible and flexible in use. In fact, data from more than 11,300 patients using *zanadio* show that a majority of patients are working and/or have children and thus have a need for an easy-to-use and flexible weight reduction program. Especially in the times of extraordinary events, such as the current COVID-19 pandemic, the availability of digital obesity treatments such as *zanadio* has become all the more significant.

Importantly, *zanadio* (and a digital health application in general) is not associated with costs for the patient (provided that the patient is covered by statutory health insurance). This differs from several other (on-site) programs for which patients have to make an advance payment and, if completing a certain number of appointments, receive a reimbursement from their health insurance company. However, individual personal and time challenges may prevent attendance at some of these mandatory appointments, which may result in non-reimbursement. The flexible use of *zanadio* can increase the adherence to the program, as suggested by preliminary findings of the clinical trial of *zanadio* (data will be published elsewhere).

### 4.2. Patient Description in the Real-World Setting

The high number of patients using *zanadio* indicates the need for a flexible and easily accessible weight reduction program. *zanadio*, as a digital weight reduction program, is very well-accepted by patients, as shown by a first clinical trial and the high number of patients with obesity that have been using the app between December 2020 and April 2022.

Interestingly, patients using *zanadio* are mainly female, despite the fact that the obesity prevalence in Germany hardly differs between sexes [13]. This finding is consistent with a recent report published by the central association of German health insurance [36] which states that 70% of digital health applications prescribed so far were prescribed to women and were, hence, mainly used by women. Moreover, it appears that women perceive obesity as a problem earlier than men and are therefore more likely to seek treatment [37,38,39]. Importantly, there are, in general, no gender-specific recommendations or guidelines for treating obesity. Weight reduction programs should, rather, be adapted to the individual situation of the patients, which primarily relates to the patient’s therapy goals, life situation, and risk profile. According to a recent review and meta-analysis [40], there currently appears to be no evidence of significantly different effects for gender-specific and gender-neutral interventions aimed at weight reduction or weight stabilization. However, further research on this topic is warranted.

Across all age groups, obesity is a major risk factor for a variety of physical comorbidities, such as type 2 diabetes, hypertension, coronary heart disease, as well as psychological disorders, including anxiety disorders and depression [2,19,20]. In line with this, patients using *zanadio* also suffer from various comorbidities, including those mentioned above; more than one-third of the patients reporting comorbidities have suffered or are currently suffering from hypertension, and nearly 18% report having diabetes (no distinction between type 1 and 2 diabetes). Such data demonstrate the need to treat obesity in order to reduce the associated risks and severity of its related diseases. In fact, studies have shown that even a moderate weight loss of 3–5% positively affects obesity-related comorbidities and improves physical and psychological comorbidities [14,21,22,41]. One major cause of type 2 diabetes is obesity. Not all patients with obesity may develop diabetes, but a large proportion of patients with diabetes are obese. Reducing excess weight is thus a main goal in treating diabetic patients with obesity and has been shown to improve physiological markers, such as the HbA1c value [42,43,44].

By implementing the German S3 guidelines to treat obesity [12] in the *zanadio* program, patients who use *zanadio* regularly are, on average, assumed to reach a clinically relevant weight loss of at least 5%. Other conservative or digitized multimodal approaches have been able to reduce excess weight in patients with obesity by −3.6% to −7.3% [10,11,14,26,27,28]. Preliminary results suggest that this goal will be reached and that other endpoints, such as weight-to-height ratio as a proxy for body fat distribution and psychological variables (quality of life and well-being) can also be improved by taking part in the *zanadio* program. Assessing changes of the severity of other comorbid diseases or changes in medication use as a proxy for disease severity may provide insight into the overall health-promoting effects of *zanadio*. Future studies should, therefore, also quantify these effects.

### 4.3. Outlook: Where to Go from Here?

The passage of the relevant laws in Germany has paved the way for digitized obesity treatments. Although *zanadio* (as any other digital health application) is supposed to be a stand-alone therapy, it does not intend to replace other treatments, such as on-site treatments. It is, rather, sought to complement other treatment options and enable as many patients as possible to take part in a weight reduction program as soon as they are seeking help. Unfortunately, while the prevalence of overweight and obesity is rising, there is still a coverage gap, at least in Germany [12]. This might be related to the fact that obesity was long considered to be a lifestyle problem rather than a disease. Only recently, awareness of obesity as a disease has grown, leading, for instance, to the endeavor to develop a structured treatment program (disease management program, DMP) in order to improve current care options for patients [45]. Using digital treatments, such as *zanadio*, might be an important piece of the DMP to provide care for as many patients with obesity as possible. In fact, for *zanadio*, it is planned to offer additional services, such as additional nutrition or exercise consultations. Also, hybrid models (e.g., using *zanadio* in combination with regular on-site training) could increase the efficacy of the treatment and enhance weight-loss maintenance—an assumption that has to be tested in clinical trials.

For now, digital health applications are only authorized to be prescribed to treat an existing medical condition. *zanadio* is thus intended to be used by patients with obesity (E66 according to ICD-10), which is defined as having a BMI of at least 30 kg/m^2^. However, it would certainly be wise to act and intervene already at an earlier stage, namely if people start becoming overweight or start developing and consolidating an unhealthy lifestyle in terms of low physical activity or inadequate nutrition. A key target group for obesity prevention is definitely children and adolescents, as overweight often starts at a young age and progressively increases over time [46,47]. Future digital treatment programs should thus target this group.

A long-term goal and important achievement for weight reduction programs would be to be able to decide in advance which patient would benefit from which type of therapy (e.g., digital, outpatient, or bariatric) or which therapeutic path within a certain treatment program. The aim of analyzing the huge data set of patients using *zanadio* in the real-world setting is to strive to identify predictors of treatment success for the digital setting (e.g., personality traits, individual barriers and motivators, or individual expectations and previous experiences), which might be assessed when patients enroll in the *zanadio* program. As a result, *zanadio* therapy could be more individually tailored to patients’ profiles and usage behavior and automatically suggest the most effective therapy path.

### 4.4. Summary

The emergence of digital health applications is a promising opportunity to close the current coverage gap for patients with obesity. So far, digital health applications are only available in Germany. Other countries will soon follow. Verification of their efficacy is still pending for most of these health applications. *zanadio*, as a guideline-based, multimodal treatment, aims to help patients with a BMI of 30–45 kg/m^2^ to develop and maintain a healthy lifestyle. The 12-month program empowers patients regarding obesity-related self-management in order to lose excess body weight, maintain that weight loss, improve physical and mental health, and reduce the risk of developing obesity-related comorbidities. The healthcare effect and the long-term maintenance of weight loss is currently being investigated in a randomized clinical trial and will be additionally tested in a real-world setting.

## Figures and Tables

**Figure 1 nutrients-14-03172-f001:**
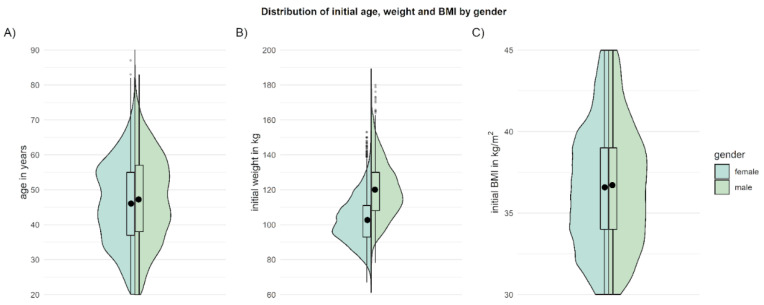
Age (**A**), initial weight (**B**), and initial BMI (**C**) of patients when registering for *zanadio*. Depicted are means (black dot) and distributions (boxplot, violin plot) separately for male and female patients.

**Figure 2 nutrients-14-03172-f002:**
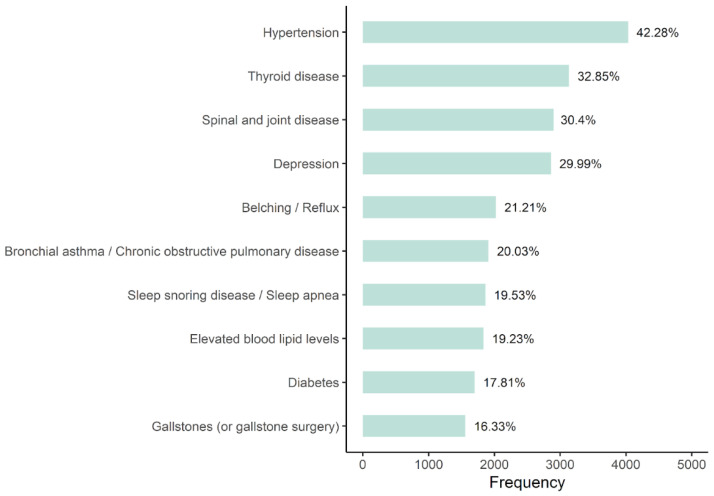
Frequencies of the ten most frequently reported comorbidities. Given are both absolute (*x*-axis) and relative values. Data of *n* = 1784 (15.8%) patients were not available.

**Table 1 nutrients-14-03172-t001:** Indication and prerequisites, contraindications and exclusion criteria for using the app *zanadio*.

Indication and Prerequisites	Contraindications and Exclusions
Patients with diagnosed obesity ICD-10: E66.- (body mass index from 30.0 to 40.0 kg/m^2^)	Body mass index outside the range of indication
Age 18 years or older	Obesity surgery (e.g., bypass, sleeve) less than 1 year ago or still lying gastric band or gastric balloon
Physical and mental ability to perform obesity treatment independently	Lack of resources for change (i.e., willingness and possibility to adapt lifestyle) and lack of motivation to undergo therapy
Need-based support for the entire duration of the program by a physician is recommended, regardless of who prescribes *zanadio* (physician, psychotherapist, health insurance company)	Acute suicidal tendencies, parasuicidal behavior
Fluency in written and spoken German	Use of compensatory measures such as vomiting, taking laxatives or diuretics, or abusive hormone use
Presence of a smartphone with iOS version 13 or higher or with Android version 5 or higher plus knowledge of how to use it	Presence of self-injurious behavior
	Suspected or existing pregnancy
	Full breastfeeding

**Table 2 nutrients-14-03172-t002:** Demographics of patients registered between November 2020 and April 2022 (BMI 30–45 kg/m^2^).

Variable	Patients, *n* (%)
Education	
University degree	1872 (16.53)
High school/university entrance diploma	1285 (11.35)
Secondary school	1171 (29.59)
Secondary school (‘Hauptschule’)	3351 (10.34)
None	55 (0.49)
Others	1628 (14.38)
Employment status	
Full-time (>35 h/week)	3890 (34.35)
Part-time (<35 h/week)	2325 (20.53)
in training	212 (1.87)
Currently not working/unemployed	528 (4.66)
Retired	872 (7.70)
Others	1537 (13.57)
Shift work	
Yes	1319 (11.65)
No	6647 (58.70)
Marital status	
Married/in partnership	7178 (63.39)
Single	1455 (12.85)
Divorced/widowed	732 (6.46)
Children	
Yes	6454 (57.00)
No	2909 (25.69)

The table gives absolute numbers of patients and frequencies relative to the number of patients that provided the respective information. Please note that 17.3% of patients did not provide information regarding education, employment status, marital status, and children. A total of 29.7% of patients did not provide information regarding shift work.

## Data Availability

Not applicable.

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
