# Peer review of "Introducing zanadio—A Digitalized, Multimodal Program to Treat Obesity"

_nutrients, 2022, doi:10.3390/nu14153172_

Round 1

Reviewer 1 Report

The manuscript titled "Introduzing zanadio - a digitalized multimodal program to treat obesity" by Forkmann et al describes new tool to treat and manage body weight. The manuscript is well written with enough details, it can be improved by introducing the following feedback:

1. Lines 25-47: Please include background on other digital tools that have been introduced to manage body weight, specifically comparing the outcomes with previously introduced digital tools and why zanadio is novel

2. Lines 196-198: Please explain the rationale behind the hypothesis that at least 5% weight loss will be observed with zanadio

3. It will be a stronger manuscript if the results are presented and demonstrate the effectiveness of zanadio in either reducing body weight or maintaining a healthy weight

Reviewer 2 Report

The authors state that the article is aimed at outlining the methodological aspects of a clinical study recently conducted to demonstrate the positive health effects of zanadio.

In this regard, the authors also state that data acquisition of the active phase of the study has been finalized and the data has been analyzed; nonetheless, the results of this analysis are not presented nor discussed in the article and will be published later. Therefore, it is not yet possible for the authors to demonstrate the positive health effects of zanadio.

Moreover, the authors state that the article is also aimed at introducing zanadio, its aims, functionality and therapeutic concept. I think that, regarding functionality, the article does not make a relevant research contribution such as a software architecture.

In summarizing, the research contribution of the work is not clear nor sufficiently relevant yet to justify the publication of the article in its present form. I think that the authors should consider resubmitting the article.

Reviewer 3 Report

The zanadio product sounds like a valuable tool for helping individuals to lose weight and help manage and/or reduce the risk for obesity-associated chronic conditions. However, I have many concerns about the manuscript.

The first is the level of industry involvement, specifically oversight of statistical analysis. This introduces a strong risk of bias into the manuscript.

In addition, much of the language use sounds boastful with a marketing tone. For a peer-reviewed publication, the language needs to be just the facts.

The inclusion of the protocol for the clinical trial before the overview of the findings of an unrelated report of those currently using the application is confusing.

I suggest rethinking this manuscript. Focus on the zanadio as a weight loss tool and provide the overview of the patients currently using this product. In discussion/conclusion, add the need for a clinical trial to confirm efficacy.  Also, hire editor with peer-review manuscript experience to strip out promotional language and ensure just the facts are being reported. Then in second publication, report the methods and findings of the clinical trial.

Lines 31, 40, 43, 45, 52, 55, 81, 122, 127, 130: Add source

Line 56: It is not clear to me what is meant by temporarily and locally restricted. Please clarify

Line 62: “current guidelines” – whose?

Lines 71-77:

Line 72: increase in calorie demand, perhaps change to increase demand for nutrient-dense calories

Line 118: Table 1 is missing from manuscript

Line 147-148: Delete PI information

Line 152: Outcomes and Measures: evaluation of behavioral change features (lines 75-77) not included

Line 176: Table 2, remove bullets.  In exclusion criteria, what do you mean by “lack of change resources”? Suggest explaining this is narrative above.

Lines 188-189: Control group  offered access to zanadio 12 months post-trial. This can be a limitation in the study design because it may influence the behavior of the control group. Need to discuss. Also, the ability of the control group to opt to try other products and weight loss strategies muddies the data findings.

Line 201: Why are behavioral change indicators not included as secondary endpoints?

Line 329: Discussion, add strengths and limitations. Limitations include incentive, offer for control group to use product 12-months post-study, lack of standardized control group protocol, involvement of industry in data analysis.

Round 2

Reviewer 2 Report

I think that the information about the aims and therapeutic concept of the mhealth app zanadio and the demographic information about the patients that are using this tool is not sufficiently relevant to justify the publication of the article.

In fact, I believe that it is still too early to publish the results of the research, which are only preliminary results. The information presented in this manuscript should be published along with the protocol and the results of the clinical trial in a future publication because the achievement of the objectives of the work cannot be currently measured.

Reviewer 3 Report

The manuscript reads much better and most of my initial concerns are mostly addressed. There are still some edits that I feel are required before approving for publication.

Line 34-35: It is the first country worldwide to introduce digital health applications (abbreviated 34 ‘DiGA’) – I tried to find support for this statement in the literature and was not successful.  Recommend changing to:

It is the first country worldwide to introduced digital health applications (abbreviated 34 ‘DiGA’) in December 2019 after passing two laws, the Digital Healthcare Act (DVG) and

Line 52: Delete link to the product website in parentheses: (https://diga.bfarm.de/de/verzeichnis/294, available online on 28.06.2022). Instead add to reference list and assign a citation number to include in the manuscript.

Lines 52-62. Since this international journal, please add examples with a global perspective. Specifically, products that are available in countries beyond Germany.

Line 62: Change from three parts to two parts.

Line 65: Suggest changing “first impression” to “initial findings.”

Lines 136-137: Same comment as line 52 above re citation.

Line 159: First inclusion criterium is listed two times. Delete second instance. Also, in inclusion/exclusion chart, be consistent with initial capitalization, e.g., change “need based” to “Need based…”

Line 343-344: I don’t believe this is an accurate statement: “So far, digital health applications are only 343 available in Germany. Other countries will follow soon.”  Perhaps it is true for zanadio?  As a general class, there are abundant digital health applications available in the US. Some offered through specific health plans in partnership with commercial entities—a model similar to what you describe with zanadio.
